# Remediation of Cadmium and Lead in Mine Soil by Ameliorants and Its Impact on Maize (*Zea mays* L.) Cultivation

Qiyue Chen, Lei Wang, Bo Li, Siteng He, Yang Li, Yongmei He, Xinran Liang * and Fangdong Zhan

College of Resources and Environment, Yunnan Agricultural University, Kunming 650201, China; 15187169949@163.com (Q.C.); wanglei.ko@foxmail.com (L.W.); ecolibo@126.com (B.L.); sitenghe@ynau.edu.cn (S.H.); liyang0125@hotmail.com (Y.L.); eyongmei06@126.com (Y.H.); zfd97@ynau.edu.cn (F.Z.)
* Correspondence: lxr8900@live.com

**Abstract:** The soil in a lead–zinc mining area, contaminated with heavy metals like cadmium (Cd) and lead (Pb), poses a risk to crops such as maize. Experiments using biochar and sepiolite as soil ameliorants in potted maize showed these substances can mitigate heavy metal contamination. Biochar increased potassium and phosphorus in the soil and maize, while sepiolite significantly boosted overground phosphorus by 73.2%. Both ameliorants transformed Cd and Pb into a more stable state in the soil, reducing their accumulation in maize, especially with biochar, which effectively inhibited metal migration during leaching events. This study provided insights for further improvement of soil amendments and multi-factor application experiments.

**Keywords:** cadmium; lead; interflow; maize; sepiolite; biochar

## 1. Introduction

The problem of heavy metal contamination in soil has become increasingly serious in recent years [1]. Cadmium (Cd) and lead (Pb) contamination of soils around the world has become a priority environmental problem [2]. Anthropogenic activities such as mining, smelting, chemical production, and factory emissions lead to the release of large amounts of Cd and Pb into the environment and cause widespread soil contamination [3]. According to the National Soil Pollution Survey Report released by the Ministry of Environmental Protection and the Ministry of Land and Resources, 16.1% of sampling sites exceeded the environmental quality standards in agricultural soils [4]. Among these agricultural land sites with contamination exceeding the standards, heavy metals and inorganic pollutants account for 82.8% of all compounds in excess of the standards [5]. Specifically, the amounts by which Cd and Pb exceed the maximum allowable concentrations in soil are 7.0% and 1.5%, respectively [4]. These heavy metals mainly accumulate in the surface layer of farmland and are easily absorbed by crops [6]. Furthermore, under rainfall conditions, Cd and Pb can be lost through surface runoff and subsurface flows, which are important pathways for the migration of Cd and Pb from agricultural soils to surrounding environments [7].

Yunnan Province in China is rich in mineral resources, particularly with extensive reserves of lead and zinc deposits [8]. The Lanping zinc–lead mine is a representative example of a large-scale zinc–lead mine in Yunnan. It covers an area of over 6.9 square kilometers and contains more than 15 million tons of Pb and Zn reserves. It is one of the largest zinc–lead mines in the world. Numerous studies have consistently shown significant accumulations of heavy metals in the soil surrounding the mining area, as well as increased heavy metal content in plant tissues, where Cd and Pb act as major pollutants. Furthermore, this mining region has a pronounced monsoon climate characterized by dry winters and rainy summers, with rainfall concentrated between June and October [9]. Consequently, agricultural lands adjacent to the mining areas experience significant surface runoff and leaching losses, exacerbating the problems of Cd and Pb contamination in agricultural soils.

In addition, maize cultivation dominates as the main crop in this region because it can survive excessive amounts of heavy metals such as Cd and Pb. Therefore, it is crucial to carry out soil remediation measures in this Cd and Pb-contaminated region to ensure the safety of plants for human consumption.

Currently, the primary approaches to remediating Cd and Pb-contaminated soils include chemical, physical, biological, and combined remediation techniques [10]. Chemical immobilization involves applying a certain amount of an immobilizing agent to the soil to reduce the bioavailability and mobility of heavy metals through various reactions, such as adsorption, precipitation, complexation, ion exchange, and redox reactions [11]. The aim of this approach is to achieve effective remediation of polluted soils. Chemical immobilization technology offers advantages such as simplicity and high efficiency, making it a preferred choice for large-scale remediation of agricultural land contaminated with heavy metals [12].

Commonly used fixatives include inorganic materials such as clay minerals, lime, and phosphorus-containing substances and organic materials such as biochar, organic fertilizers, and agricultural wastes [13]. Among these materials, sepiolite and biochar show the strongest passivation effect on Pb and Cd. Due to its large surface area and high ion exchange capacity, sepiolite is often used in soils contaminated with heavy metals [14]. Previous studies have shown that sepiolite exhibits effective immobilization effects on soils contaminated with heavy metals while reducing the uptake of Cd and Pb by maize. The porous structure and oxygen-rich functional groups of biochar enable efficient adsorption of heavy metals, dyes, phosphates, etc. [15–18]. Biochar can be obtained from various sources, such as through the high-temperature pyrolysis of plants or agriculture, to achieve effective utilization of waste resources [16–19]. Due to the low cost of raw materials used in the production of biochar, and its excellent pollutant removal properties, it is an economically and environmentally friendly material that has attracted attention in the scientific community. The use of biochar and sepiolite for soil remediation shows their potential as environmentally friendly materials for immobilizing heavy metals.

Research on the remediation of soils contaminated with heavy metals using ameliorants has been carried out both domestically and internationally, mainly through pot and field experiments, to study their effects on the availability of Cd and Pb in soil, as well as the accumulation of these metals by plants. However, only limited studies have been conducted on the influence of ameliorants on the leaching loss of Cd and Pb. Therefore, in this study, farmland soil contaminated with heavy metals from a lead–zinc mine in Jinding City, Lanping County, Yunnan Province, was used as the experimental soil. Biochar and sepiolite were selected as passivation materials for conducting indoor experiments with maize cultivation. The primary aims were to investigate the effectiveness of biochar and sepiolite in immobilizing Cd and Pb in the soil, the uptake of these metals by maize plants, and their influence on the leaching loss of Cd and Pb from the soil. The other goal of this research is to provide valuable insights into the remediation of arable soils contaminated with Cd and Pb as well as effective measures to control their migration into soils.

## 2. Materials and Methods

### 2.1. Experimental Materials

The pot experiment was conducted in the greenhouse of the Scientific Research Center around Back Mountain at Yunnan Agricultural University. The soil used was collected from lead and zinc mine-polluted farmland located in Jinding Town, Yunnan Province, where the Field Test Center is situated. The soil was naturally dried, purified by removing plant roots and other impurities, and sieved through a 20-mesh screen (2 mm). The type of fertilizer used was a compound fertilizer specifically for maize. The fertilizer application rate was calculated at 0.33 g per kg of soil, based on the standard of applying 50 kg of fertilizer for every 150,000 kg of soil. The basic physical and chemical properties of the soil were as follows: pH 5.83, organic matter carbon content 29.15 g/kg, total nitrogen 1.82 g/kg, total phosphorus 0.89 g/kg, total potassium 20.54 g/kg, alkali-hydrolytic nitrogen 271.02 mg/kg, available phosphorus 47.10 mg/kg, and available potassium

171.27 mg/kg. The total Cd concentration was found to be 3.6 mg/kg, while the total Pb concentration was 68.24 mg/kg [20].

### 2.2. Experimental Design

The PVC pipe diameter of the leaching device is 11 cm, with a height of 55 cm. The actual soil filling is 40 cm, weighing 2.5 kg. Leaching solutions were collected at depths of 10 cm, 20 cm, and 30 cm from the bottom layer. Biochar and pumice stone were separately applied to the topsoil (0–20 cm) in agricultural fields. A layer of quartz sand measuring 2–3 cm was placed on top of the soil, while a layer measuring 3–4 cm was placed underneath to prevent water infiltration and drainage from affecting the soil. The quartz sand was soaked in nitric acid before being laid down. The leaching solution used amounts of 25% of the annual rainfall in field conditions [21], which means that 6 L was applied to the potted plants each time.

The passivators used in the experiment were straw biochar (prepared by pyrolyzing rice straw at 500 °C for 10 h under oxygen-limited conditions) and seafoam, which were purchased from Henan Lize Environmental Protection Technology Co. The pores of sepiolite consisted of mesopores (3.41 nm < pore width < 50 nm) and macropores (50 nm < pore width < 86.31 nm). The pores of biochar consisted of micropores (0.89 nm < pore width < 2 nm) and mesopores (2 nm < pore width < 33.24 nm) [20]. The BET surface area of biochar is 204.73 $m^2/g$ [20]. CK indicate no passivator added to soil. BC indicate biochar added to soil. SP indicate seafoam added to soil.

Seeds that were uniform in size and fully mature were selected for planting after undergoing surface sterilization with a solution containing $H_2O_2$ at a concentration of 10%. After rinsing with distilled water, the seeds were placed on moist filter paper in Petri dishes and incubated at a temperature of 25 °C for three days until germination. Two maize seedlings were retained during the seedling stage.

Soil leachate and infiltrate were collected twice, with a 1-day interval, 45 days after maize planting. The water-soluble concentrations of Cd and Pb in the leachate at depths of 10 cm, 20 cm, 30 cm, and 40 cm were quantified. The Cd and Pb contents in potted soil as well as the levels of N, P, and K were determined. Furthermore, the cadmium and lead concentrations in the overground and belowground plant tissues along with the N, P, and K levels in field-grown maize were assessed.

### 2.3. Determination of N, P, and K Elements in Soil

The determination of total nitrogen in the soil was conducted using the semi-micro Kjeldahl method. The acid dissolution-molybdenum antimony colorimetric method was employed for the determination of total phosphorus in soil. The flame photometer method was used for measuring total potassium in soil. The diffusion–absorption method after alkaline hydrolysis was utilized for determining mineralized nitrogen in soil. The 0.5 mol/L sodium bicarbonate extraction–molybdenum antimony colorimetric method was applied to measure available phosphorus in soil, while the 1 mol/L ammonium acetate extraction-flame photometer method was used for determining available potassium in soil [22].

### 2.4. BCR Sequential Extraction of Cd and Pb Content and Available Cd and Pb Content in Soil

Soil available Cd and Pb were extracted using a 0.01 mol/L CaCl$_2$ extraction solution. The specific method was as follows: 2.0 g of soil sample was weighed and placed in a 50 mL centrifuge tube, followed by the addition of 20 mL of 0.01 mol/L CaCl$_2$ solution (liquid-to-soil ratio of 10:1). This mixture was shaken at 25 °C and 180 r/min for 2 h and then centrifuged at 3600 r/min for 5 min. The filtrate was collected and analyzed using flame atomic absorption spectrophotometry [23].

Soil Cd and Pb forms were determined using the BCR sequential extraction method [24].

### 2.5. Determination of Cd and Pb Content in Soil Flow

Leachate was collected from soil at depths of 10 cm, 20 cm, and 30 cm, as well as runoff water at a depth of 40 cm. The samples for Cd and Pb analysis were digested using the nitric acid—peroxide method and measured using a Graphite Furnace Atomic Absorption Spectrometer [25].

### 2.6. Various Indicators Were Measured after Maize Growth for 45 Days

Plant height was determined using conventional methods, while the dry weight was measured after drying at 100 °C and weighing after drying at 80 °C to deactivate the chlorophyll. Due to the limited biomass of maize samples, overground and underground parts were separately analyzed for nutrient content using the $H_2SO_4$-$H_2O_2$ digestion method. Total nitrogen was quantified by colorimetric analysis with Nessler's reagent; total phosphorus was determined by the molybdenum antimony anti-reagent colorimetric method; and total potassium content was measured using flame photometry [26]. For Cd and Pb analysis in maize samples, they underwent digestion with an $HNO_3$-$HClO_4$ solution followed by measurement via flame atomic absorption spectrophotometry [27].

## 3. Results

### 3.1. The Impact of Applying Biochar and Sepiolite on the Physicochemical Properties of Soil

After the application of biochar and sepiolite, the physicochemical properties of the soil are shown in Figure 1. There was no significant change in soil pH after the application of biochar ($p < 0.05$). In addition, the introduction of seafoam significantly increased the soil pH by 11.01% (Figure 1a). This indicates that sepiolite can effectively alleviate soil acidification.

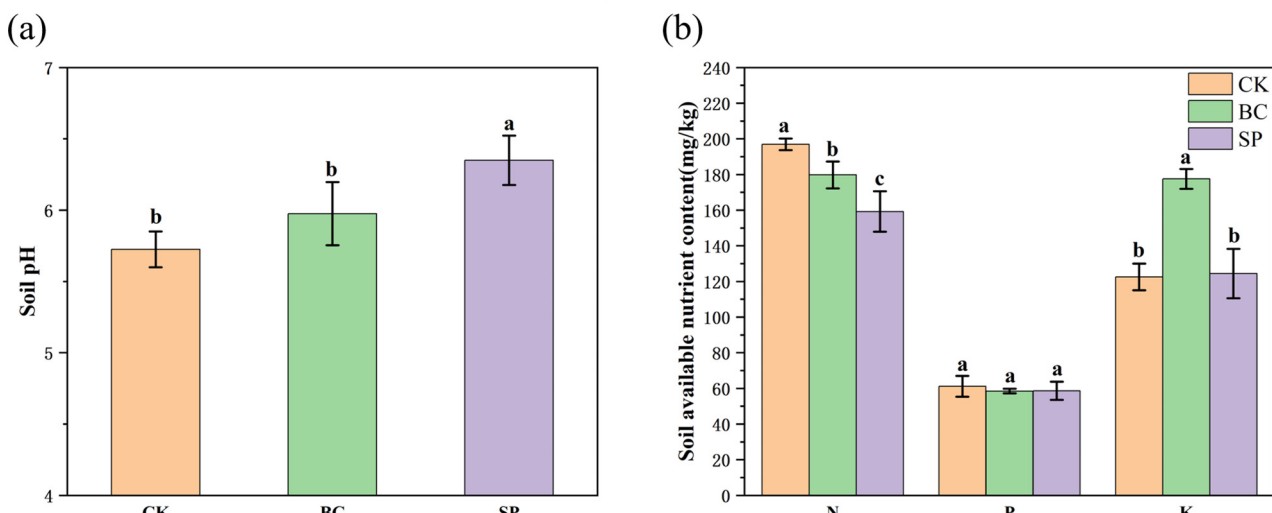

**Figure 1.** pH (**a**) and available N, P, and K (**b**) under different ameliorant treatments. CK: no passivator added to soil, BC: biochar added to soil, SP: seafoam added to soil. Data were means ± SD (*n* = 3). The bars indicate the standard error of the mean. Column with the same letter has no significant differences between treatments at *p* < 0.05 according to the least significant difference (LSD) test.

The impact of the application of a soil conditioner on nutrient content is shown in Figure 1. Application of biochar and seafoam to the soil significantly reduced the available nitrogen content of the soil by 8.98% and 19.11%, respectively. Compared to soils without added ameliorants, the changes in available phosphorus in soils after the application of biochar and sepiolite are not significant ($p < 0.05$). Meanwhile, compared to soils without added ameliorants and those with added sepiolite, the content of available K in soils significantly increases after the application of biochar ($p > 0.05$) (Figure 1b). This may be because biochar comes from agricultural residues and is naturally rich in nutrients.

### 3.2. The Impact of Applying Biochar and Sepiolite on the Growth and Nutrient Uptake of Maize

The physicochemical properties of maize plants changed after the application of biochar and sepiolite (Figure 2). After the application of biochar, the reduction in the overground and underground biomass of maize plants is not significant ($p < 0.05$). After the application of sepiolite to the soil, the change in the overground biomass of maize plants is not significant ($p < 0.05$) (Figure 2a). These results indicate that the application of ameliorants to the soil had little effect on the growth of maize.

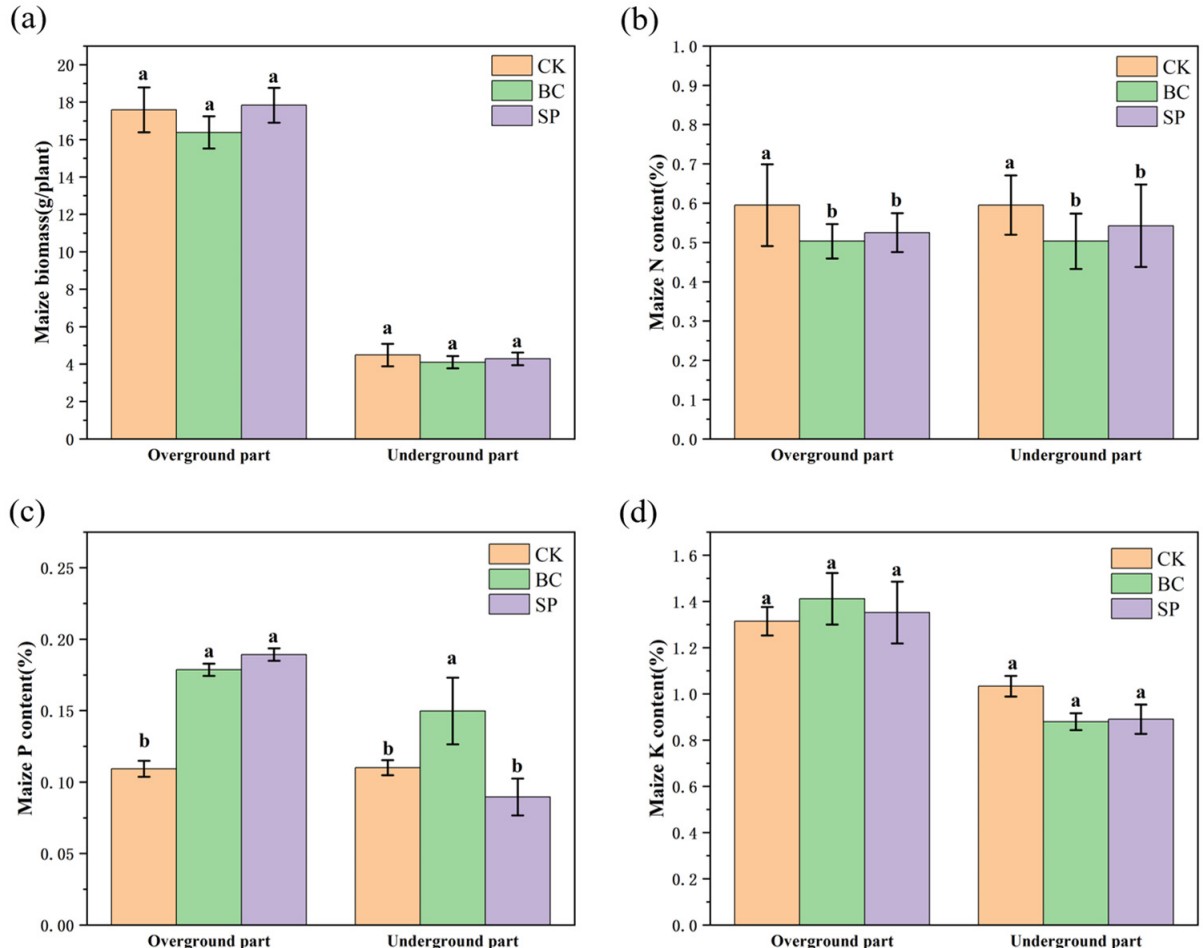

**Figure 2.** The biomass (**a**), N (**b**), P (**c**), and K (**d**) content in maize under different ameliorant treatments. CK: no passivator added to soil, BC: biochar added to soil, SP: seafoam added to soil. Data were means $\pm$ SD ($n$ = 3). The bars indicate the standard error of the mean. Column with the same letter has no significant differences between treatments at $p < 0.05$ according to the least significant difference (LSD) test.

Further analysis of N, P, and K contents of aboveground and belowground parts of maize showed that there was no significant difference in N uptake between aboveground and belowground parts of maize by application of biochar or seafoam (Figure 2b). Biochar application significantly increased the P content in the above-ground and below-ground parts of maize by 63.64% and 36.36%, respectively. The P content of the aboveground part of maize was significantly increased by 72.73% after the application of seafoam (Figure 2c). There was no significant difference in K uptake between aboveground and belowground parts of maize by application of biochar or seafoam (Figure 2d). In conclusion, the application of biochar effectively increased P uptake in maize plants, while the addition of seafoam was beneficial in promoting P uptake in the aboveground part of maize.

### 3.3. The Impact of Applying Biochar and Sepiolite on Different States of Cd and Pb in Soil

After applying sepiolite and biochar to the soil, the changes in the contents of extracted Cd and Pb are shown in Figure 3a. The addition of biochar and seafoam to the soil significantly reduced the extracted cadmium content of the soil by 32.67% and 28.71%, respectively. The application of the two ameliorants did not significantly affect the content of extracted Pb in the soil (Figure 3a).

(a)

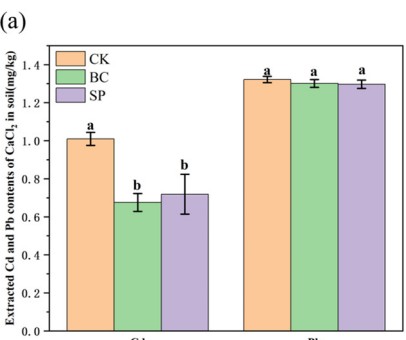

(b)

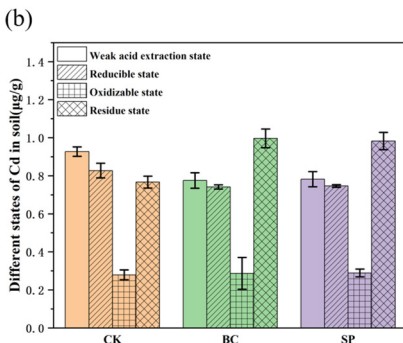

(c)

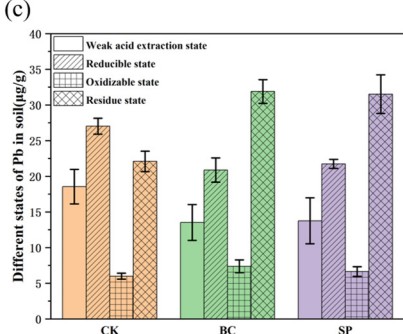

**Figure 3.** Extracted Cd and Pb (**a**); different states of Cd (**b**) and Pb (**c**) under different ameliorant treatments. CK: no passivator added to soil, BC: biochar added to soil, SP: seafoam added to soil. Data were means ± SD ($n$ = 3). The bars indicate the standard error of the mean. Column with the same letter has no significant differences between treatments at $p < 0.05$ according to the least significant difference (LSD) test.

By further employing BCR sequential extraction on Cd and Pb in the soil, the changes in the content of various metal states are shown in Figure 3b. Application of seafoam and biochar reduced the weakly acid-extracted cadmium content of the soil by 16.40% and 15.86%, respectively. Meanwhile, the more stable residual cadmium content in the soil increased by 29.87% and 27.27%, respectively (Figure 3b). There was no significant change in the content of the oxidized state or reduced state of Cd in the soil after the application of the ameliorants. These results indicate that the application of ameliorants can transform the more active weak acid-extractable form of Cd in the soil into a stable residual state, thereby reducing its biological availability.

The application of seafoam and biochar reduced the weakly acidic extractable lead content of the soil by 27.12% and 25.88%, respectively. At the same time, the more stable residual lead content in the soil increased by 44.36% and 42.69%, respectively. The reduced Pb content in the soil decreased by 22.76% and 19.58%, respectively, compared to the decrease in the Pb content in only the weakly acid-extractable state (Figure 3c). These results indicate that the application of ameliorants can transform the more active weakly acid-extractable and reduced states of Pb in the soil into a stable residual state, thereby reducing its biological availability.

### 3.4. The Impact of Applying Biochar and Sepiolite on the Accumulation of Cd and Pb in Maize

To further confirm the changes in the available Cd and Pb contents in the soil after the application of biochar and ameliorants, we measured the variations in the overground and underground Cd and Pb contents in the planted maize (Figure 4). The results are shown in the figure. The addition of biochar and sepiolite resulted in a significant reduction in the accumulation of Cd in the underground parts of maize, with decreases of 51.2% and 49.8%, respectively (Figure 4a). The application of ameliorants did not lead to a significant improvement in the overground accumulation of Cd in maize. The addition of biochar led to a significant reduction in both the overground and underground accumulation of Pb in maize, with decreases of 37.8% and 53.1%, respectively. The application of sepiolite to the soil resulted in decreases of 11.7% and 27.1% in the overground and underground accumulation of Pb in maize, respectively (Figure 4b). These results indicate that the remediation effects of the two ameliorants on different states of Cd and Pb in the soil are

similar, and biochar exhibits a more significant inhibitory effect on the accumulation of Pb in maize. This may be related to the differences in the material properties of biochar and sepiolite.

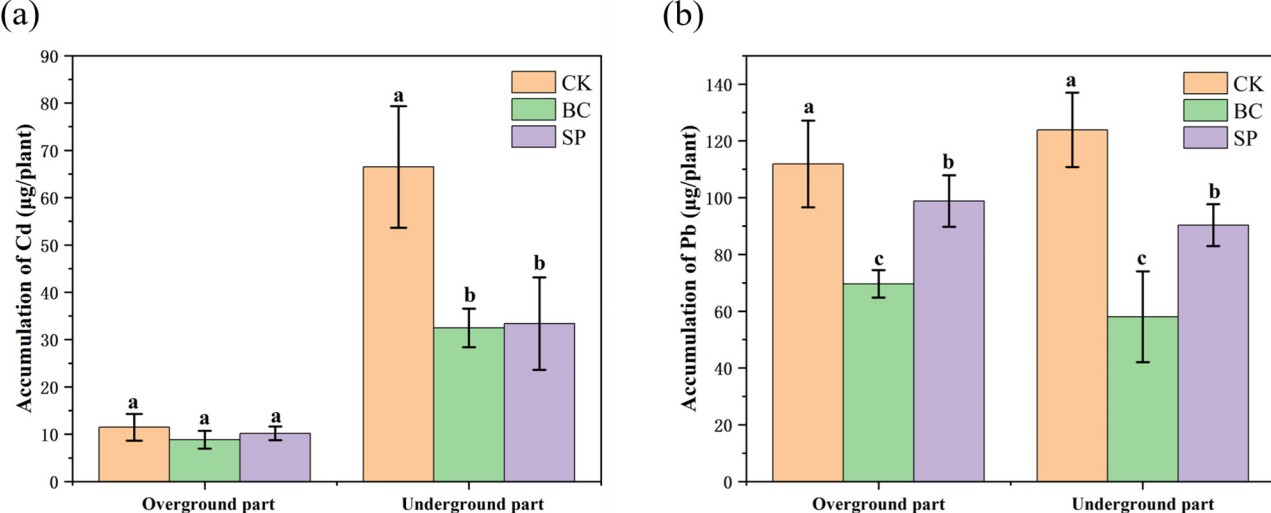

**Figure 4.** The accumulation of Cd (**a**) and Pb (**b**) in the overground and underground parts of maize under different ameliorant treatments. CK: no passivator added to soil, BC: biochar added to soil, SP: seafoam added to soil. Data were means ± SD (*n* = 3). The bars indicate the standard error of the mean. Column with the same letter has no significant differences between treatments at *p* < 0.05 according to the least significant difference (LSD) test.

*3.5. The Impact of Applying Biochar and Sepiolite on the Concentrations of Cd and Pb in the Leachate at Different Soil Depths*

In this study, simulated rainfall was applied twice to leach the potted soil (Figure 5). The alterations in Cd and Pb concentrations in the leachate at various depths (10 cm, 20 cm, 30 cm, and 40 cm) of the potted soil were measured after each leaching (Figure 6). In all leaching experiments, the leaching amounts of Cd and Pb in the soil leachate increased with depth. The addition of biochar and sepiolite both resulted in a significant decrease in the leaching concentration of Cd in the first soil leachate, with reductions of 38.3% and 71.2%, respectively. The addition of biochar led to a significant decrease in the leaching concentration of Cd in the second soil leachate, with a reduction ranging from 28.8% to 40.5%. Furthermore, through a comparison of the Cd content in the leachate at different depths, it was found that the increase in Cd in the leachate above 30 cm was the most significant (Figure 6a). This may be attributed to the fact that Cd is mostly enriched in the surface layers of the soil.

In the initial leaching, the reduction in the leachate concentration of Pb after the application of biochar decreased from 43.17% to 32.60% with increasing depth. After the application of sepiolite, the reduction in leachate concentration of Pb decreased from 31.81% to 16.26% with increasing depth. In the second leaching, the decrease in the leachate concentration of lead (Pb) following the application of biochar enhanced significantly, rising from 32.75% to 62.39% as the depth increased. After the application of sepiolite, the reduction in leachate concentration of Pb increased from 10.51% to 27.02% as the depth increased (Figure 6b). These results indicate that biochar can more effectively inhibit the migration of Cd and Pb than sepiolite in the soil under two consecutive leaching events. Furthermore, under rainfall conditions, Pb is more mobile in soil than Cd.

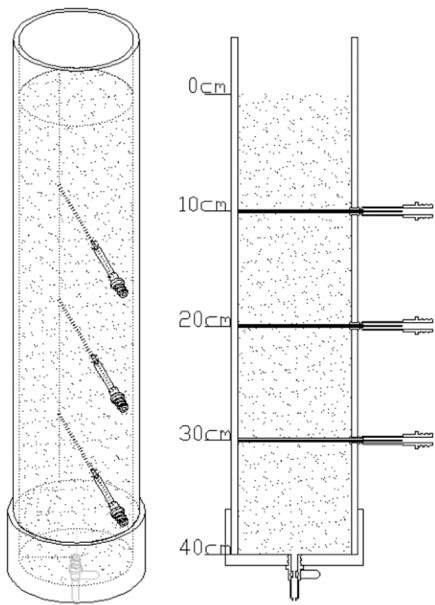

**Figure 5.** Schematic Diagram of Leaching Experiment Apparatus.

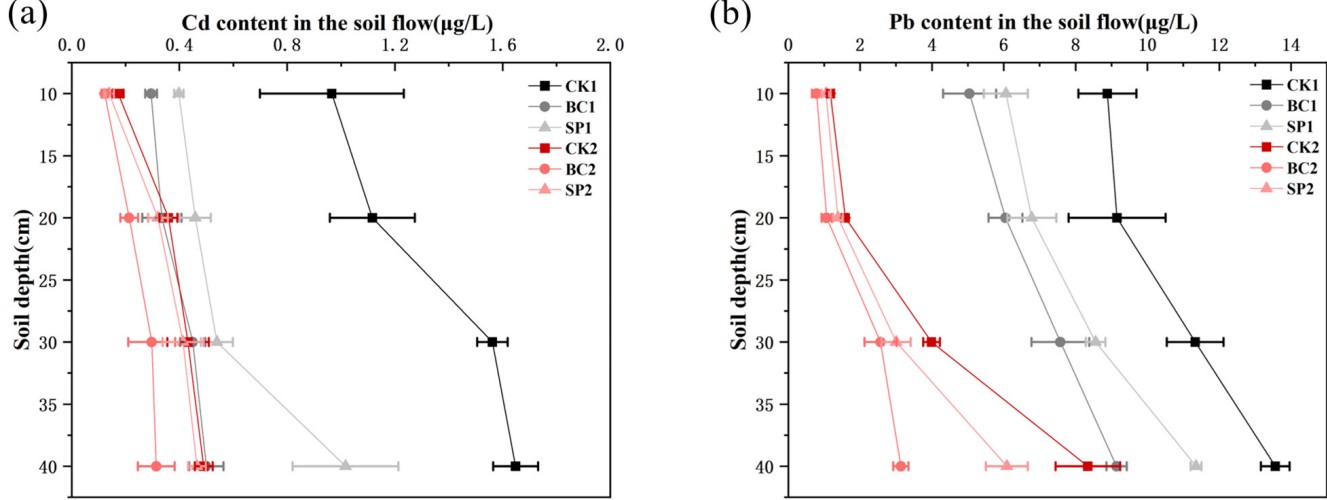

**Figure 6.** The concentration changes of Cd (**a**) and Pb (**b**) in soil leachate at different depths under two leaching events. CK: no passivator added to soil, BC: biochar added to soil, SP: seafoam added to soil. Serial number 1 is the first leaching, serial number 2 is the second leaching. Data were means ± SD (*n* = 3).

## 4. Discussion

In summary, using biochar and sepiolite as amendments in soils affected by mining significantly reduces the availability of Cd and Pb. This results in lower accumulation of these heavy metals in maize grown on contaminated soils. The addition of biochar and sepiolite leads to a slight increase in soil pH, a substantial decrease in the weak acid-extractable forms of Cd and Pb, and an increase in the stable residual state of these metals. Additional leaching experiments showed that biochar and sepiolite additives effectively reduce the mobility of Cd and Pb in soil water flow. Furthermore, during two simulated rainfall events, the amounts of leached Cd and Pb from the soil progressively decreased with each leaching cycle. The application of biochar and sepiolite leads to the deprotonation of soil surfaces due to the presence of hydroxyl groups on the passivator, thereby raising soil pH [28]. Consequently, applying these additives to mining-contaminated soils can effectively transform Cd and Pb from a weakly acid-extractable state to a more stable

residual form, thereby diminishing their mobility in the soil. This method is significantly beneficial for remediating heavy metal pollution in mining soils and simultaneously reduces the accumulation of these metals in maize.

In leaching experiments, soil pH and Cd and Pb in the leachate were affected by the application of ameliorants. The increase in the soil pH may be due to the direct effect of the applied amendments. However, the reduction in the Cd and Pb in the leachate contents might be attributed to two factors: (1) Cd and Pb were leached out during the first and second leaching intervals, and (2) improved metal immobilization by the applied additives [29].

The findings of this study demonstrate that the implementation of ameliorants effectively mitigates metal availability in mining soil, as evidenced by the measured soil pH value and Cd concentration in the leaching solution at the conclusion of the experiment. An elevation in soil pH is instrumental in remediating contaminated soils, primarily because most metals become less mobile under alkaline conditions, thereby decreasing their bioavailability [30]. Furthermore, this rise in soil pH enhances the negative charge, which in turn promotes the precipitation and adsorption of metals onto the soil surface [31]. Additionally, the introduction of ameliorants not only alters the soil's physicochemical properties but also effectively reduces the availability of cadmium (Cd) and lead (Pb) by inducing modifications in the soil's structure [32]. This finding also confirms that the actual hazardous metal influences are dependent on not only their total contents but also their available concentration in soils.

The dissolved weak acid-extractable state of heavy metals has the highest mobility and bioavailability, facilitating the absorption and utilization of heavy metals by plants [33]. It is the primary source of heavy metals in plant tissues. Therefore, the transformation of highly biologically available weak acid-extracted heavy metals into less utilizable residual forms is crucial for soil metal immobilization [34]. The speciation of heavy metals in the soil determines their effectiveness and content in plant tissue. Immobilization of metals reduces the uptake of heavy metals by plants, thereby reducing their accumulation in crops and minimizing potential health risks associated with metal transfer through the food chain [15,35]. In this study, the proportion of residual Cd and Pb was significantly increased by the addition of biochar. Furthermore, the addition of biochar resulted in a significant reduction in belowground Cd and Pb levels as well as their cumulative amounts in maize plants. Similarly, the addition of sepiolite resulted in a significant reduction in subsurface Cd and Pb contents as well as their cumulative amounts. These results are consistent with previous studies by Wu et al. (2017) and Luo et al. (2014) [36,37].

Sepiolite has a significant specific surface area and an abundant microporous structure and it effectively immobilizes heavy metals on its rough surface through isomorphic substitution and surface complexation reactions. It stably adsorbs Cd and Pb in its inner layer and simultaneously forms unstable outer layer complexes [14,38]. Biochar, with its well-developed pore structure and large surface area, can selectively adsorb heavy metal elements in the soil through competitive adsorption and cation-$\pi$ interactions. This helps inhibit the migration of heavy metal elements [39,40]. Moreover, previous studies have indicated that the addition of biochar improves the composition and stability of water-stable aggregates in the soil while also expanding the pore space in the soil. This further enhances the reaction between abundant adsorption sites on pore walls with heavy metal ions to increase the adsorption capacity of the soil [41,42]. The results of this study showed that biochar and sepiolite significantly reduced soil Cd loss during the two leaching experiments. Furthermore, biochar significantly reduced soil Pb loss in the leaching experiments.

The characteristics of different heavy metal elements in the soil also greatly affect their mobility after passivation. Previous studies have indicated that Cd generally has a weaker adsorption affinity for soil particles than Pb [43]. Cd may be more mobile than Pb in acidic soils, where it can form more soluble complexes [44]. The ionic charge of metal ions influences their interactions with soil particles [45]. Cd is commonly found as $Cd^{2+}$, which can be more mobile due to its smaller size and higher charge compared to $Pb^{2+}$. Therefore,

although the total Pb concentration in potted soil was nearly 20 times higher than the total Cd concentration, the effective concentration of Pb was only 0.7 mg/kg higher than the effective concentration of Cd. The underground accumulation of Pb in maize was only 58.33% higher than the accumulation of Cd. After the application of the soil ameliorant, Cd in the soil could adsorb onto the ameliorant more quickly than Pb. Therefore, in the two leaching experiments, Cd in the soil was immobilized by the soil ameliorant, resulting in less Cd leaching as the depth increased compared to when the ameliorant was not added. In contrast, Pb in the soil, not having migrated to the ameliorant location, showed a noticeable increase in leachate Pb content with increasing depth.

Additionally, it is noteworthy that the availability of heavy metals in soil can be influenced by the application of passivates, which induces changes in soil texture [46]. It was reported that clay soil can retain more metals than sandy soil. Soils containing complex oxygenated compounds and clay contents can exchange and adsorb more ions [47,48] The fine particles in sepiolite and biochar add a certain sand quality to the soil, and the application of sepiolite and biochar can increase the permeability of the soil and improve its texture [49–51]. Therefore, in this study, the application of sepiolite and biochar was likely to promote the migration and fixation of heavy metal ions to the surface of the passivator by enhancing the change in soil permeability. These processes and mechanisms need to be further explored.

## 5. Conclusions

This study conducted greenhouse experiments on the application of biochar and sepiolite for the remediation of heavy metal-contaminated soil. This research aims to investigate the effects of introducing passivation agents on heavy metal accumulation in maize cultivated in polluted soil. The results indicate that the application of ameliorants can transform the more active weak acid-extractable state of Cd and Pb in the soil into a stable residual state, thereby reducing its biological availability. The addition of biochar and sepiolite resulted in a significant reduction in the accumulation of Cd in the underground parts of maize, with decreases of 51.2% and 49.8%, respectively. The addition of biochar led to a significant reduction in both the overground and underground accumulation of Pb in maize, with decreases of 37.8% and 53.1%, respectively. Biochar more effectively inhibited the migration of Cd and Pb than sepiolite in the soil under two consecutive leaching events. Furthermore, under rainfall conditions, Pb was more mobile in the soil than Cd. This study reveals the vertical migration patterns of Cd and Pb in contaminated soil during rainfall after the application of ameliorants. It offers valuable insights for the application of soil ameliorants in agricultural land contaminated with Cd and Pb, and also sheds light on crop cultivation strategies that can be effectively implemented during soil remediation processes. The ameliorant mechanisms of Cd and Pb on sepiolite and biochar require further study.

**Author Contributions:** Validation, L.W.; Formal Analysis, S.H.; Resources, B.L.; Writing—Original Draft, Q.C. and Y.L.; Writing—Review and Editing, Y.H., X.L., and F.Z. All authors have read and agreed to the published version of the manuscript.

**Funding:** The authors thank the Youth Project of Yunnan Provincial Basic Research Program (No. 202201AU070178), the National Natural Science Foundation of China (42177381 and 42267002), the expert workstation of Longhua Wu in Yunnan Province (202305AF150042), and Yunnan Agricultural University Research Initiation Project (No. F2022-07) for financial support of this research.

**Data Availability Statement:** The original contributions presented in the study are included in the article.

**Conflicts of Interest:** The authors declare no conflicts of interest.

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
