# Peer review of "Remediation of Cadmium and Lead in Mine Soil by Ameliorants and Its Impact on Maize (Zea mays L.) Cultivation"

_agronomy, doi:10.3390/agronomy14020372_

Round 1
Reviewer 1 Report
Comments and Suggestions for Authors
Dear authors,
The manuscript “Remediation of Cadmium and Lead in Mine Soil by 2 Ameliorants and Its Impact on Maize (Zea mays L.) Cultivation” deals with the soil in a lead-zinc mining area, contaminated with heavy metals (Cd and Pb), poses a risk to the maize. The goals of the research are: 1) to study the effectiveness of biochar and sepiolite in immobilizing Cd and Pb in the soil, in the uptaking of these metals by maize plants and in the leaching loss of Cd and Pb from the soil; 2) to test a method for restoring arable soils contaminated with Cd and Pb. A pot experiment using biochar and sepiolite as soil ameliorants was conducted in the greenhouse of the Research Center at Yunnan Agricultural University.
The topic of the article is quite relevant, especially considering the increasing soil contamination by heavy metals where Cd and Pb may act as major pollutants. The research was done correctly. The pot experiments with maize using biochar and sepiolite as soil ameliorants were conducted. The pot experiments showed these ameliorants can mitigate studied heavy metal contamination.The authors present interesting material. However, statistical processing of the results was not performed, making it impossible to assess the significance of the differences in the experiment's variants. The results are presented as 196.88±3.25. It is unclear whether this represents the mean ± standard deviation or ± mean error.
The authors should revise the presentation of the results and resubmit the manuscript to the journal.

Author Response
Dear Reviewer
Thank you very much for your comments. We have put the reply in the attachment.
Sincerely,
Xinran Liang

Reviewer 2 Report
Comments and Suggestions for Authors
Remediation of Cadmium and Lead in Mine Soil by 2 Ameliorants and Its Impact on Maize (Zea mays L.) Cultivation
General comment:
The study investigates the impact of biochar and sepiolite as soil ameliorants in a lead-zinc mining area with heavy metal contamination, specifically cadmium (Cd) and lead (Pb), and its risk to crops like maize. The findings offer insights into the use of soil ameliorants for remediation in contaminated farmlands and crop cultivation. The results presented interesting. Though this manuscript has some issues, which needs clarification. Some of results have not been critical reviewed with respect to previous studies.
Specific comments:
Introduction
1. Give more details on how Cd and Pb metal leaching occurs in soil and the importance relates to the present study.
2. You mentioned that the soil contamination rate in China is 16.1%. How this is comparable with the acceptable levels?
3. Explain the passivation in terms of heavy metal immobilization.
4. Give 1-2 sentences about how biochar is synthesized and its uses like remediation of a range of contaminants (heavy metals, dyes, phosphates, etc) in soil and water. Also, explain why it is economically beneficial and famous among the scientific community. The following are good papers. Refer them and cite them.
a. https://link.springer.com/article/10.1007/s42773-023-00212-2
b. https://www.sciencedirect.com/science/article/pii/S0013935120311804
c. https://www.sciencedirect.com/science/article/pii/S2214714423008978
5. How biochars work on immobilizing heavy metals. How do biochar properties affect metal migration?
Materials and methods
6. You never mentioned the biochar type and sepiolite used for your experiments? Did you make biochar in the lab or it’s a commercial one?
7. In supplementary information, please add the following:
i. Why the surface sterilization of seeds with a solution containing H2O2. Is essential?
ii. Briefly mention the significance of collecting data at different depths and the potential implications for understanding metal mobility in the soil.
8. Replace “graphite furnace” with “Graphite Furnace Atomic Absorption Spectrometer”. You used GFAAS to measure Cd, and Pb content in soils and flame AAS to measure the Pb, and Cd content in digested maize plants. Any difference?
9. The significance of using the BCR Sequential Extraction Method
10. “limited biomass of maize plants”. How does this affect the interpretations of the study?
11. Provide any diagrams of the experimental setup including the PVC pipe leaching device.
Results
Section 3.1
12. Define CK, BC and SP.
13. What is the benefit of increasing soil pH after applying biochar and sepiolite? Have any previous studies reported similar enhancements?
14. The paragraph mentions a decrease in available nitrogen and phosphorus and an increase in available potassium after applying biochar and sepiolite into soil samples. What nutrients/elements in agricultural biochar and sepiolite are responsible for this? Give some references.
15. Have you conducted the BET surface area measurements of sepiolite and the biochar? If possible, add them or cite them from relevant sources.
Section 3.2
16. Check “increased” or “decreased”: Line 176 After the application of biochar to the soil, the aboveground and underground biomass of maize plants increased and decreased from 17.58±1.19 g/plant and 4.48±0.60 g/plant to 16.38±0.86 g/plant and 4.1±0.32 g/plant, respectively.
17. What is the benefit of section 3.2? Explain the significance or implications of these changes Iine 185-196. Why two ameliorants have different effects on P absorption might enhance comprehension.
Section 3.3
18. Add cross reference. When biochar and sepiolite were added to the soil, the content of available Cd in the soil decreased from 1.01±0.03 mg/kg to 0.68±0.05 mg/kg and 0.72±0.1 mg/kg, respectively. “ Some cross-references (e.g., Figure 3a) are missing in paragraph and in the entire manuscript. Please include that.
19. Is this Pb or Cd? Line 217 “Meanwhile, the more stable residual Cd content in the soil increased from 22.09±1.43 μg/g to 31.89±1.66 μg/g and 31.52±2.70 μg/g.’ Check the whole section.
20. Explain briefly
a. The importance of decrease of weakly acid-extractable Pb content. If possible, add any mechanisms (adsorption, precipitation, complexation, etc.) behind the transformation of metal states.
b. the importance of reducing the biological availability of Cd and Pb.
Section 3.4
21. Two ameliorants reduced the Cd accumulation of underground parts of maize. Is it significant?
22. Briefly explain why biochar has more pronounced inhibitory activity in terms of Pb accumulation over sepiolite. Also, you mentioned that” This may be related to the differences in the material properties of biochar and sepiolite.” What material properties of biochar?
Section 3.5
23. Why biochar is more effective in controlling metal migration over sepiolite
24. Explain why “furthermore, under rainfall conditions, Pb is more mobile in soil than Cd.’
25. Any similar studies reported the utilization of biochar and sepiolite for heavy metal immobilization? Tabulate.
Discussion
26. What I have noticed is that you have separated the discussion from results and discussion. Please combine them as shown in the agronomy journal. Refer some papers there.
27. Some plagiarized sections were detected in page 8. Rearrange it.
Conclusion
28. Not conclusion It should be conclusions. Add “s”
29. Highlight the significance of the study and any limitations.
Author contributions
· Replace X,X,Y,Y.. etc with relevant initials of authors
Provide funding details
Grammar
· Correct the grammar mistakes (e.g., missing punctuations, missing hyphens), throughout the manuscript. Improve the word efficiency. For example,
o The aim of this research is- Replace with “this research aims”
Comments on the Quality of English Language
Correct the grammar mistakes (e.g., missing punctuations, missing hyphens), throughout the manuscript. Improve the word efficiency. For example,
o The aim of this research is- Replace with “this research aims”
Author Response
Dear Reviewer
Thank you very much for your comments. After careful modification, we put the reply in the attachment.
Sincerely,
Xinran Liang

Round 2
Reviewer 1 Report
Comments and Suggestions for Authors
Review 2
Dear authors,
You have corrected the deficiencies identified by the reviewers. Unfortunately, the analysis of the results still contains errors and inaccuracies.
I recommend authors not only to correct the text according to the reviewers' comments, but also to carefully check how consistently the results are presented, whether there is any confusion in the order of discussion of the results obtained, etc.
Lines 173-174
“The available phosphorus content also showed a slight 173 decrease from 61.21±5.86 mg/kg to 58.50±1.27 mg/kg and 58.66±5.12 mg/kg, respectively”.
It is not correct!
Mobile phosphorus differences were insignificant (Figure 2).
Lines 175-176
“Meanwhile, the content of available K increased from 122.51±7.5 mg/kg to 177.51±5.56 175 mg/kg and 124.41±13.82 mg/kg, respectively”
It is not correct!
Mobile phosphorus has difference only in BC.
Lines 177-179
The increase in porosity after biochar application is not really confirmed in this paper. Was this increase statistically significant? This sentence should probably be deleted, or reference should be made to literature data confirming an increase in porosity when similar doses of biochar are applied to the same (similar) soil.
Lines 181-183
Should these two sentences be moved to the next section?
Lines 188-190
After the application of biochar to the soil, the aboveground 188 and underground biomass of maize plants decreased… ???
Figure 2 a shows what these differences in biomass are insignificant.
The same notes are for others – check please the correction of you description.
Line 216
Figure 3a is the copy of the figure 1 a. Why? Is it correct to repeat the same information? Is it mistake?
And so on…
Author Response
Dear reviewer
Thank you very much for your questions and suggestions to help us improve the quality of manuscript. We have carefully made responses and revisions to related issues, and have placed these responses and revisions in the attachments.
Sincerely
Xinran Liang

Reviewer 2 Report
Comments and Suggestions for Authors
Here is an ideal paper to get the surface area for biochar synthesized from Henan Lize. They have reported that biochar's surface area is 26 m2/g. https://link.springer.com/article/10.1007/s42773-023-00220-2
But, the paper itself doesn't mention the type of biochar. You may need to get an idea by contacting Henan Lize.
If this is for rice straw biochar, then you can add the value and cite the paper.
But, please note it is not a high surface biochar.
Author Response
Dear reviewer
Thank you very much for your questions and suggestions to help us improve the quality of manuscript.
Response: Thank you very much for your wonderful suggestion. The Biochar which was use in the article “Biochar addition to tea garden soils: effects on tea fluoride uptake and accumulation” was produced by slow pyrolysis of miscellaneous wood materials at 500 ℃ under anoxic conditions for 2–3 h and provided by Henan Lize Environmental Protection Technology Co., Ltd (Zhengzhou, China). This may be due to the differences in the properties of the rice straw biochar we used. We used the same type and batch of rice straw biochar as mentioned in the article “Field experiment on the effects of sepiolite and biochar on the remediation of Cd- and Pb-polluted farmlands around a Pb–Zn mine in Yunnan Province, China”, where the specific surface area is reported as 204.73 m2/g. Once again, thank you very much for your suggestion. Please refer to lines 112 to 113.
Sincerely
Xinran Liang

Round 3
Reviewer 1 Report
Comments and Suggestions for Authors
Dear authors, you have made improvements to the narrative in your manuscript. However, I must point out that there are still some shortcomings. Here below are a few examples of this. I have only given some examples of errors and unclear descriptions of the work done. The manuscript is not yet considered ready for publication.
Lines 9-11
Biochar significantly increased potassium and phosphorus in the soil and maize, while sepiolite boosted overground phosphorus by 73.2%.
The proposal doesn't make sense. It is difficult to understand the quantitative meaning of the term "significant". Next: Is 73.2% a significant increase or not?
Lines 13-14.
It is hardly possible to recommend the use of the studied ameliorants on contaminated agricultural land after a single experiment.
Lines 23-25.
…contamination rate in China is 16.1%
What does it mean? Is it the percentage of annual land pollution? The total land area of the country? Or just agricultural land? Or is it an increase in the level of contamination, i.e. an increase in the average level of contaminants in all soils in China?
Lines 25-26.
The meaning of this sentence is also unclear.
Lines 26-27.
7.0% and 1.5% are compared to what figures? Average content of elements in soil-forming rocks? Or is it an excess over the maximum allowable concentrations?
Lines 33-34.
The Lanping LEAD–ZINC MINE is a representative ex-33 ample of a large-scale LEAD–ZINC MINE in Yunnan.
Lines 37-38.
Why does a lead-zinc mine cause Cd and Pb pollution? And what is about Zn?
Lines 90-92
The meaning of this sentence is unclear. The meaning of this phrase is unclear. Where did these fertilizer rates come from? What kind of fertilizer was used?
Lines 92-96.
Organic matter content. What do the authors mean? Organic matter content or organic carbon content? What methods were used to determine all these parameters? To what extent are the maximum allowable concentrations of Cd and Pb exceeded?
And so on.
Author Response
Dear reviewer
Thank you for your suggestions.
Lines 9-11
Biochar significantly increased potassium and phosphorus in the soil and maize, while sepiolite boosted overground phosphorus by 73.2%.
The proposal doesn't make sense. It is difficult to understand the quantitative meaning of the term "significant". Next: Is 73.2% a significant increase or not?
Response: Thank you for your suggestions. After consideration, the term 'Significant' indeed appears abrupt here, so we have removed 'Significant'. Moreover, through subsequent data analysis, there is a significant increase of 73.2%. Please refer to lines 9-11.
Lines 13-14.
It is hardly possible to recommend the use of the studied ameliorants on contaminated agricultural land after a single experiment.
Response: Thank you for your suggestions. It is difficult to determine the universality of passivator effects in complex soil and agricultural planting processes through single-factor experiments. Therefore, we will further explore by conducting multifactorial experiments based on single-factor experiments, which will include modified amendments, amendments compound formulations, and combinations with planting methods. And we change the last sentence to “This study provided insights for further improvement of soil amendments and multi-factor application experiments.” Please refer to lines 13-14.
Lines 23-25.
…contamination rate in China is 16.1%
What does it mean? Is it the percentage of annual land pollution? The total land area of the country? Or just agricultural land? Or is it an increase in the level of contamination, i.e. an increase in the average level of contaminants in all soils in China?
Response: Thanks for your question. In the agricultural land sampling sites of the national soil survey, 16.1% of the sampling sites have soil contamination exceeding the standards. Please refer to lines 24-25.
Lines 25-26.
The meaning of this sentence is also unclear.
Response: Thanks for your question. We change “among them” to “Among these agricultural land sites with contamination exceeding the standards”。 Please refer to line 25-26.
Lines 26-27.
7.0% and 1.5% are compared to what figures? Average content of elements in soil-forming rocks? Or is it an excess over the maximum allowable concentrations?
Response: Thank you for your suggestions. This is an excess over the maximum allowable concentrations. And we change this sentence to “Specifically, the amounts by which Cd and Pb exceed the maximum allowable concentrations in soil are 7.0% and 1.5%, respectively.” Please refer to line 27-29.
Lines 33-34.
The Lanping LEAD–ZINC MINE is a representative ex-33 ample of a large-scale LEAD–ZINC MINE in Yunnan.
Response: Thank you for your suggestions. We have made modifications to the capitalization. Please refer to line 35-36.
Lines 37-38.
Why does a lead-zinc mine cause Cd and Pb pollution? And what is about Zn?
Response: Thank you for your questions. Due to mining activities in lead-zinc mines, tailings piles leak large amounts of Cd and Pb, and the toxicity of these two elements is much higher than that of zinc. On the other hand, referencing the regulations of the 'Soil Environmental Quality Standard' (GB 15618-2018), the soil Cd content in the area exceeds the maximum allowable value by three times, Pb is close to the maximum allowable value, while the Zn content is 154.8 mg/kg, which is below the maximum allowable concentration of 200 mg/kg.[1]
Reference
Zhan, F.; Zeng, W.; Yuan, X.; Li, B.; Li, T.; Zu, Y.; Jiang, M.; Li, Y. Field experiment on the effects of sepiolite and biochar on the remediation of Cd- and Pb-polluted farmlands around a Pb-Zn mine in Yunnan Province, China. Environ Sci Pollut Res Int. 2019, 26, 7743-7751. doi: 10.1007/s11356-018-04079-w
Lines 90-92
The meaning of this sentence is unclear. The meaning of this phrase is unclear. Where did these fertilizer rates come from? What kind of fertilizer was used?
Response: Thank you for your suggestions. This sentence was changed to “The fertilizer application rate was calculated at 0.33 g per kg of soil, based on the standard of applying 50 kg of fertilizer for every 150,000 kg of soil”. The type of fertilizer used was a compound fertilizer specifically for corn. Please refer to lines 92-95.
Lines 92-96.
Organic matter content. What do the authors mean? Organic matter content or organic carbon content? What methods were used to determine all these parameters? To what extent are the maximum allowable concentrations of Cd and Pb exceeded?
Response: Thank you for your questions. The organic matter content is organic matter carbon content. We change it in line 96. These data are based on previous studies of the soil in the area, and relevant literature from previous research has been added. Please refer to line 100. In China, the maximum allowable concentrations of Cd and Pb in agricultural soil, according to the 'Soil Environmental Quality Standard' (GB 15618-2018), are: 0.3 mg/kg for Cd, and 70 mg/kg for Pb. Therefore, this article selects soil where Cd exceeds the standard by 12 times, while Pb is close to the maximum allowable concentration.
